# A Benchmark of
# Medical Out of Distribution Detection

**Tianshi Cao**
Vector Institute, University of Toronto

**Chin-Wei Huang**
Mila, University of Montreal

**David Yu-Tung Hui**
Mila, University of Montreal

**Joseph Paul Cohen**
Mila, University of Montreal

## Abstract

**Motivation:** Deep learning models deployed on medical tasks can be equipped with Out-of-Distribution Detection (OoDD) methods in order to avoid erroneous predictions. However it is unclear which OoDD methods are effective in practice.

**Specific Problem:** Systems trained for one particular domain of images cannot be expected to perform accurately on images of a different domain. These images should be flagged by an OoDD method prior to prediction.

**Our approach:** This paper defines 3 categories of OoD examples and benchmarks popular OoDD methods in three domains of medical imaging: chest X-ray, fundus imaging, and histology slides.

**Results:** Our experiments show that despite methods yielding good results on some categories of out-of-distribution samples, they fail to recognize images close to the training distribution.

**Conclusion:** We find a simple binary classifier on the feature representation has the best accuracy and AUPRC on average. Users of diagnostic tools which employ these OoDD methods should still remain vigilant that images very close to the training distribution yet not in it could yield unexpected results.

## 1 Introduction

A safe system for medical diagnosis should withhold diagnosis on cases outside its validated expertise [1, 2, 3]. For machine learning (ML) systems, the expertise is defined by the validation score on the distribution of data used during training, as the performance of the system can be validated on samples drawn from the same distribution (as per PAC learning [4]). This restriction can be translated into the task of *Out-of-Distribution Detection* (OoDD), the goal of which is to distinguish between samples in and out of the training distribution of the diagnosis system (abbreviated to *In* and *Out* data). We consider a pipeline where the example is filtered through the OoD detector, and only examples predicted as *In* are passed to the downstream ML predictor.

In contrast to natural image analysis, medical image analysis must often deal with orientation invariance (e.g. in cell images), high variance in feature scale (in X-ray images), and locale specific features (e.g. CT) [5]. A systematic evaluation of OoDD methods for applications specific to medical image domains remains absent, leaving practitioners blind as to which OoDD methods perform well and under which circumstances. This paper fills this gap by benchmarking many OoDD methods under various medical image types. More specifically, we conduct four experiments, each on a specific medical imaging dataset as *In* data (frontal and lateral chest X-ray, fundus imaging, and histology). Each experiment includes comparisons to three categories of *Out* data taken from 14 datasets, and 21 configurations of OoDD methods. Our empirical studies show that these OoDD methods perform poorly when detecting correctly acquired images that are not represented in the training data (later called use-case 3). We also find that some simple methods such as a binary

Submitted to the 35th Conference on Neural Information Processing Systems (NeurIPS 2021) Track on Datasets and Benchmarks. Do not distribute.

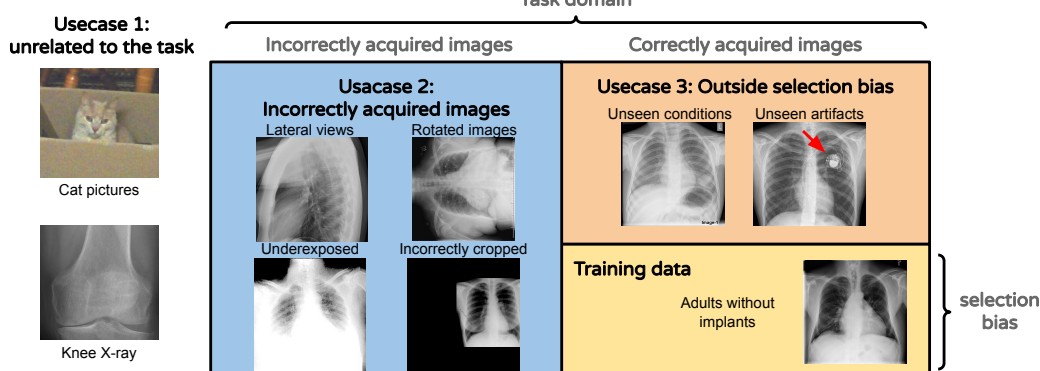

Figure 1: The three use-cases shown in relation to each other. The training data is sampled iid from the *In* data distribution. 1) Inputs that are unrelated to the task. 2) Inputs which are incorrectly prepared 3) Inputs that are unseen due to a selection bias in the training distribution.

classifier on features trained on *In* data performed on par with more complex methods (see Figure 4). We hope that this work can inspire more discussion and future work on the unique challenges of OoDD in medical image domains.

## 2    Defining OoD in Medical Data

Given an *In* distribution dataset, how should we define what constitutes *Out* data? To address this, we identify three distinct out-of-distribution categories:

- **use-case 1** Reject inputs that are unrelated to the evaluation. This includes obviously-wrong images from a different domain (e.g. MRI images processed using a model trained on X-ray images) and less obviously-wrong images (e.g. wrist X-ray image processed using a model trained with chest X-rays).
- **use-case 2** Reject inputs which are incorrectly prepared. For example, in the case of chest X-ray images: blurry images, poor contrast, incorrect view of the anatomy (lateral views processed using a model trained with frontal views), images with the incorrect file format or pre-processing applied), or changes in data acquisition protocol.
- **use-case 3** Reject inputs that are unseen due to a selection bias in training data (e.g. image with an unseen disease or underrepresented demographic), which may yield unexpected results.

We justify these use-cases by enumerating different types of mistakes or biases that can occur at different stages of the data acquisition. This is visually represented in Figure 1. Note that earlier use-cases take precedence over later ones, such that if an input meets the definition of use-case 1 OoD, it falls under use-case 1 and we do not need to consider whether it's also incorrectly prepared. We construct our experiments to evaluate OoDD methods' performance on each category. We specifically include use-case 1 as a sanity check and for completeness, as the OoD methods should work here. Systems can be deployed in settings with natural images. A hospital PACS (Picture Archiving and Communication System) may have debugging or phantom images that the model should not make predictions for.

**Example 1**    As running example, we will use our first evaluation where the *In* data consists of frontal chest X-rays. The *In* data contains 10 pulmonary conditions in the NIH ChestX-ray14 dataset [6]. In use-case 1 we include natural images, images of symbols and text, and skeletal X-ray images. Use-case 2 contains lateral view chest x-rays. Finally, use-case 3 include frontal chest X-rays of four pulmonary conditions that were not present in *In* data.

## 3    Task Formulation

In this paper, we will either assume that the downstream task is to perform classification using a deep neural network, which we call the task network, or use an auxiliary model designed specifically for OoD detection.

For the auxiliary models, we use the same in-distribution set (i.e. the training set) to train the auxiliary model as the one used to train the classifier. This is done so that the auxiliary model is representing the same distribution that the classifier was trained on.

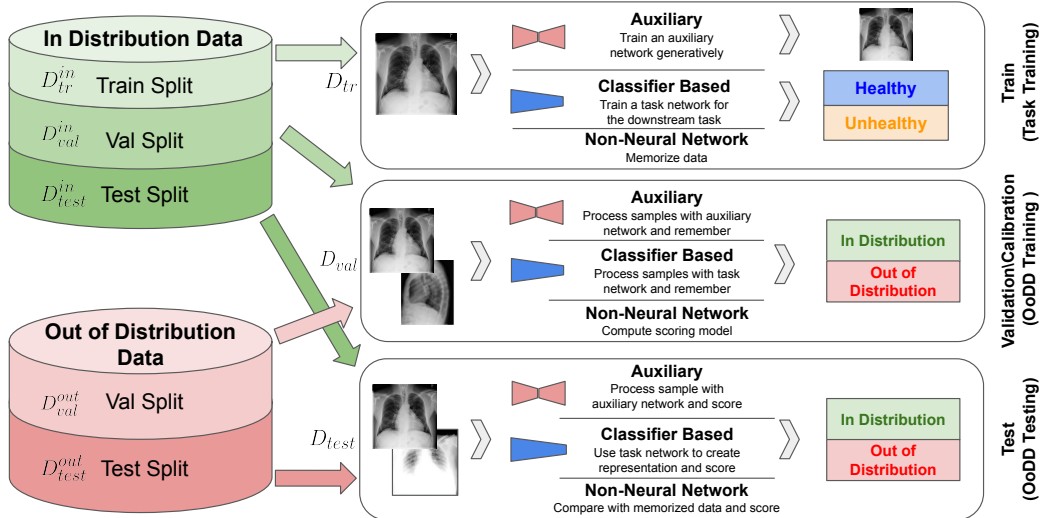

Figure 2: An overview of one experiment which is repeated for multiple seeds. Interplay of In and Out Data with three steps of OoDD evaluation. The data splits are shown on the left for the *In* and *Out* data. On the right, three parts of the evaluation are shown. At the top the classifier or auxiliary network is trained. The OoDD method is trained using calibration data in the middle and then evaluated on test data in the bottom. Also note how data is used differently in different types of OoDD methods.

To eliminate the bias of our evaluation, we randomize the choice of the in-distribution set (i.e. the training set) as well as the datasets in the calibration (validation) set and the test set, by choosing a random subset of out-of-distribution datasets for calibration and using the rest for test reporting.

For each random trial, we use the same data splitting for all models (classifier-based and auxiliary models alike). We found that certain models are more sensitive to the calibration set (which is used for threshold calibration, for example) than the training split. We believe this is due the limited number of validation datasets we are using, for deployment we would prefer to use as many as possible, but for this evaluation this can expose differences between methods. To reduce the variance, we conducted 10 trials to average out the randomness of the data splitting and report the mean and the standard error.

For test evaluation, we compute the accuracy and AUPRC on each test set, and then take the average across them all. So the imbalance due to the varying dataset sizes is not an issue.

**Notation:** Let us denote a sample of *In* data used to train the task network as $D_{tr}$. Then, an OoDD method $M$ is trained on a "calibration set" $D_{val} = D_{val}^{in} \cup D_{val}^{out}$, a union of *In* and *Out* samples (labeled as "in" or "out"). $M$ may also use the features learned by the task network, thereby also making use of $D_{tr}$. Finally, $M$ is evaluated on the test set $D_{test} = D_{test}^{in} \cup D_{test}^{out}$, also composed of *In* and *Out* samples. Each tuple $(M, D_{tr}, D_{val}^{in}, D_{val}^{out}, D_{test}^{in}, D_{test}^{out})$ constitutes an *experiment*. This three step process is illustrated in Figure 2.

## 3.1 Methods of OoDD ($M$)

We consider three classes of OoDD methods. Classifier-only methods assume access to a downstream classifier trained for classification on *In* data ($D_{tr}$). Methods with auxiliary models requires pre-training of a neural network that on *In* data using other objectives such as image reconstruction. We also consider a KNN-based approach that doesn't require training of neural networks.

**Classifier-only methods** Classifier-only methods make use of the downstream classifier for performing OoDD. Compared to data-only methods they require less storage, however their applicability is constrained to cases with classification as downstream tasks.

- *Probability Threshold* [7] uses a threshold on the prediction confidence of the classifier to perform OoDD.
- *Score SVM* [8] trains an SVM on the logits of the classifier as features, generalizing probability threshold.

- *Binary Classifier* trains on the features of the penultimate layer of the classifier. This is equivalent to attaching a binary prediction head on the classifier backbone for OoDD. The classification head is trained with SGD while weights of the backbone are kept fixed.
- *Feature KNN* uses the same features as the binary classifier, but constructs a KNN classifier in place of logistic regression.
- *ODIN* [9] is a probability threshold method that preprocesses the input by taking a gradient step of the input image to increase the difference between the *In* and *Out* data. A threshold is applied on prediction confidence to discriminate between *In* and *Out* data.
- *Mahalanobis* [10, 11] models *In* data in the feature space of the classifier with a mixture of Gaussians. To perform OoDD, images are first preprocessed through gradient steping as in ODIN, and then their feature representations are computed. Likelihood of each image is computed using the feature's weighted Mahalanobis distance to the mixture of Gaussians. Threshold on the likelihood gives prediction for OoDD. The "Mahalanobis" method concatenates the output of every dense block to get feature representations of the images, while "Single layer Maha." uses the penultimate layer outputs.

**Methods with auxiliary models**   OoDD methods in this section require an auxiliary model trained on *In* data. This results in extra setup time and resources when the downstream classifier is readily available. However, this could also be advantageous when the downstream task is not classification (such as regression) where methods may be difficult to adapt. *Autoencoder Reconstruction* thresholds the reconstruction loss of the autoencoder to achieve OOD detection. Intuitively, the autoencoder is only optimized for reconstructing *In* data, and hence reconstruction quality of *Out* data is expected to be poor due to the bottleneck in the autoencoder [12, 13, 14]. We consider three variants of autoencoders: standard autoencoder (AE) trained with reconstruction loss only, variational autoencoder trained with a variational lower bound (VAE) [15], and decoder+encoder trained with an adversarial loss such as ALI [16] or BiGAN [17]. Furthermore, we include two different reconstruction loss functions in the benchmark: mean-squared error (MSE) and binary cross entropy (BCE). Finally, *AE KNN* [18] constructs a KNN classifier on the features output by the encoder.

**Non-neural-network methods**   We also compare against KNN which is a strong simple baseline that does not utilize neural networks to construct features. This method memorizes samples from $D_{tr}$ to form a k-nearest neighbour (KNN) model, and then uses $D_{val}$ to learn a SVM using the distances to the K nearest neighbours as features. In the KNN-1 case, this SVM distills down to a single parameter representing the threshold. The SVM in KNN-8 uses 8 parameters to construct a classifier where each parameter acts as a weighting over neighboring samples ordered by proximity.

**Example 1 (cont.)**   We will use *Autoencoder Reconstruction with VAE trained using MSE Loss* (Reconst. VAEMASE) as the OoDD method of our running example. In the first stage, we train the auxiliary VAE on $D_{tr}$ by maximizing the evidence lower bound (ELBO) under MSE criteria as evidence. Then, in the second stage, we compute the reconstruction loss on samples of $D_{val}$ and calibrate a threshold value on reconstruction loss for separating *In* and *Out* samples. Finally, we evaluate on $D_{test}$ by predicting its label ("in" or "out") according to the reconstruction loss and comparing to the ground truth.

### 3.2   Description of Datasets

The following datasets are used in **use-case 1** (UC-1) Common which will be introduced in the next section:

- **MNIST** [19] 28x28 black and white hand written digits data. The original test split is used.
- **notMNIST**[1] Letters A-J in various fonts. Black and white with resolution of 28x28. The original test split is used.
- **CIFAR10 and CIFAR100** [20] 32x32 natural images. The original test split used.
- **TinyImagenet**[2] 96x96 downsampled subset of ILSVRC2012. The original validation split used.
- **FashionMNIST** [20] Grayscale 28x28 images of clothes and shoes. The original validation split is used.
- **STL-10** [21] Natural image dataset of size 96x96. 8000 testing images are used.

---

[1]http://yaroslavvb.blogspot.com/2011/09/notmnist-dataset.html
[2]https://tiny-imagenet.herokuapp.com/

155 • **Noise** White noise generated between 0 and 1 at any desired resolution.

156 The following medical imaging datasets are used:

157 • **ANHIR** [22] Automatic Non-rigid Histological Image Registration Challenge. Microscopy
158    images of histopathology tissue samples stained with different dyes. 9000 images of intestine and
159    9000 images of kidney tissue were used in evaluation 4, use-case 2.

160 • **DRD** [23] 35k retina images from 17k patients with diabetic retinopathy. Each image is labeled
161    on a scale of 0 to 4. We convert this into a classification task where 0 corresponds to healthy and
162    1-4 corresponds to unhealthy.

163 • **DRIMDB** [24] Fundus images of various qualities labeled as good/bad/outlier. This dataset is
164    specifically designed for quality assessment of images. There are 91 images labeled as bad/outlier,
165    which we use in evaluation 3, use-case 2.

166 • **IDC** [25, 26] Whole slide images of Invasive Ductal Carcinoma (IDC) tissue regions for diagnos-
167    ing breast cancer. The samples are H&E stained and estrogen receptor positive (ER+). 277,524
168    crops of 50x50 RGB images are obtained from 162 slides.

169 • **Malaria** [27] 27,558 images of cells in blood smear microscopy collected from healthy persons
170    and patients with malaria; used in evaluation 4 use-case 1.

171 • **MURA** [28] MUsculoskeletal RAdiographs is a large dataset (40,561 images total) of skeletal
172    X-rays. We use its validation split in evaluation 1 and 2's use-case 1. Images are grayscale and
173    the square cropped.

174 • **NIH Chest** [6] The NIH ChestX-ray14 Dataset is comprised of 112,120 X-ray images with 14
175    condition labels. The x-rays images are in frontal view.

176 • **PadChest** [29] This is a large scale chest X-ray dataset comprised of 160k images from 67k
177    patients, labeled with 117 radiological findings - we use the subset with correspondence to the
178    14 condition labels in the NIH Chest dataset. Images are in 5 different views: posterior-anterior
179    (PA), anterior-posterior (AP), lateral, AP horizontal, and pediatric.

180 • **PCAM** [30] The Patch Camelyon consists of 327,680 color images (96x96) extracted from
181    histopathologic scans of lymph node sections from the Camelyon dataset [31]. Images are labeled
182    for presence of cancerous tissue.

183 • **RIGA** [32] Fundus imaging dataset for glaucoma analysis. It contains 460 images annotated by
184    physicians for regions of disease. We use this dataset for evaluation 3, use-case 3.

185 ## 3.3 *In* Datasets ($D_{tr}, D_{val}^{in}, D_{test}^{in}$)

| Domain | Eval | *In* data | use-case 1 *Out* data | use-case 2 *Out* data | use-case 3 *Out* data |
|---|---|---|---|---|---|
| Chest X-ray | 1 | NIH (*In* split) | UC-1 Common MURA | PC-Lateral, PC-PED | NIH-Cardiomegaly, NIH-Nodule, NIH-Mass, NIH-Pneumothorax |
| | 2 | PC-Lateral (*In* split) | UC-1 Common MURA | PC-AP, PC-PED, PC-AP-Horizontal, PC-PA | PC-Cardiomegaly, PC-Nodule, PC-Mass, PC-Pneumothorax |
| Fundus Imaging | 3 | DRD | UC-1 Common | DRIMDB | RIGA |
| Histology | 4 | PCAM | UC-1 Common, Malaria | ANHIR, IDC | None |

Table 1: Datasets used in evaluations. UC-1 Common includes datasets such as MNIST, CIFAR-10, and random noise. PC=PadChest, NIH=NIH ChestX-ray14, DRIMDB=Diabetic Retinopathy Images Database, RIGA=Retinal fundus images for glaucoma analysis.

186 For $D_{tr}$, we select from four medical datasets ranging over three modalities of medical imaging.
187 Each dataset defines a classification task. If there are multiple independent tasks we merge them into
188 a single task because it is not clear how to deal with multiple tasks yet and the methods we evaluate
189 only expect one task. The *In* datasets of each evaluation are:

190  1. Frontal view chest X-ray images. The task is to predict if 10 of the 14 radiologcal findings
191     defined by the **NIH** ChestX-ray14 dataset [6] are present in the image. The remaining con-
192     ditions are held-out for use-case 3. The training, validation, and testing splits accompanying
193     the original data are used for $D_{tr}$, $D_{val}^{in}$, and $D_{test}^{in}$.

194  2. Lateral view chest X-ray images (PC-Lateral). The task is the same as evaluation 1, but the
195     data is from lateral view images in the PadChest (**PC**) dataset [29]. Remaining conditions
196     are also held-out for use-case 3. We randomly split the dataset in 80-10-10 ratio for $D_{tr}$,
197     $D_{val}^{in}$, and $D_{test}^{in}$.

3. Fundus/retinal (back of the eye) images. The task is to if the detect diabetic retinopathy score is $> 0$ in the retina defined by the **DRD** (Diabetic Retinopathy Detection) dataset. [23] We randomly split the original training set in 80-10-10 ratio for $D_{tr}$, $D_{val}^{in}$ and $D_{test}^{in}$. The original test set was not used due to lack of labels.

4. H&E stained histology slides of lymph nodes. The task is to predict if image patches contain cancerous tissue defined by the **PCAM** dataset [30] from the Camelyon dataset [31]. Original train, validataion, and test splits are used for $D_{tr}$, $D_{val}^{in}$, and $D_{test}^{in}$.

### 3.4 *Out* Datasets ($D_{val}^{out}$ and $D_{test}^{out}$)

We select *Out* datasets according to use-cases described in section 2. As users may be independently interested in a particular use-case, we evaluate the OoDD methods per use-case. Clearly, characteristics of each use-case are defined relative to the *In* distribution, hence we may need to select different *Out* datasets for each *In* dataset.

For $D_{val}^{out}$ and $D_{test}^{out}$ under **use-case 1**, we take a combination of natural image and symbols datasets which we call *UC-1 Common*. This is used for every *In* data. For **use-case 2**, we use datasets of the same modality of the *In* distribution, but incorrectly captured. For example, different views (e.g. lateral vs frontal) of the chest area are used as $D_{val}^{out}$ and $D_{test}^{out}$ for evaluations 1 and 2. Finally, for **use-case 3**, we use images of different conditions/diseases as *Out* data. For evaluations 1 and 2, the four held-out conditions are used as use-case 3 *Out* data. We did not include a use-case 3 *Out* dataset for histology slides due to lack of available data. Table 1 summarizes our roster of *In* and *Out* datasets. Each *Out* dataset is split 50/50 for $D_{val}^{out}$ and $D_{test}^{out}$. Subsampling is used to balance the number of *In* and *Out* samples in $D_{val}$ and $D_{test}$.

It remains to be determined how to split *Out* data between $D_{val}$ and $D_{test}$. A common but overly optimistic assumption is that *Out* data are similar to each other, hence the OoDD method is trained and evaluated on different splits of the same OoD dataset. In our running example, this entails calibrating the threshold for reconstruction loss on NIH Chest data vs MNIST training-split, and then evaluate on NIH chest data vs MNIST testing split. On the other extreme, the assumption is that we have no access to out-of-distribution data, turning the task into that of one-class classification where no *Out* data is used except for testing. In a realistic setting, the developer would train the OoDD method on a number of various datasets to cover different modes of OoD data, but the data seen at deploy time possesses variability not accounted for by those selected by the developer. Hence, for each use-case, we select a subsample of datasets for training the OoDD method, and use the remaining datasets for evaluation. In experiments where only one *Out* dataset is available, separate splits of that data is used between $D_{val}$ and $D_{test}$.

**Example 1. (cont.)**    For use-case 1 of the running example, we split the *Out* data in to 14 partitions (9 datasets in UC-1 Common, and 5 areas of the body in the MURA skeletal X-ray dataset). We sample without replacement 3 partitions for $D_{val}^{out}$, and use the rest in $D_{test}^{out}$. In use-case 2, we have lateral-view, pediatric (PED), dorsal-view (AP), and horizontal dorsal-view (AP-Horizontal) as four *Out* splits. We randomly select one as $D_{val}^{out}$ and use the remaining for $D_{test}^{out}$. We do the same for use-case 3, which also has four *Out* splits.

## 4    Experiments and Results

In this benchmark, we report the performance of each OoDD method on every evaluation and use-case averaged over 10 trials. We measure the accuracy and Area Under Precision-Recall Curve (AUPRC) on $D_{test}$, totaling at 11 pairs of performance numbers per method. Since $D_{test}$ is class-balanced, accuracy provides an unbiased representation of type I and type II errors. AUPRC characterizes the separability of *In* and *Out* samples in predicted value (the value that we threshold to obtain classification). Details of experimental setup are in Appendix A.

Figures 3, 6, 7, and 8 show the performance of OoDD methods on the four evaluations. Generally, we observe that our choice of datasets create a range of simple to hard test cases for OoDD methods. While many methods can solve use-case 1 and use-case 2 adequately in evaluations 1-3, use-case 3 proves difficult for all methods tested. This is reflected in the UMAP visualization of the AE latent spaces (column B of figures 3 to 7), in which we observe that the *In* data points are easily separable from *Out* data in use-cases 1 and 2, but well-mixed with *Out* data in use-case 3. It is surprising that no method achieved significantly better accuracy than random in use-case 3 of evaluations 1 and 2

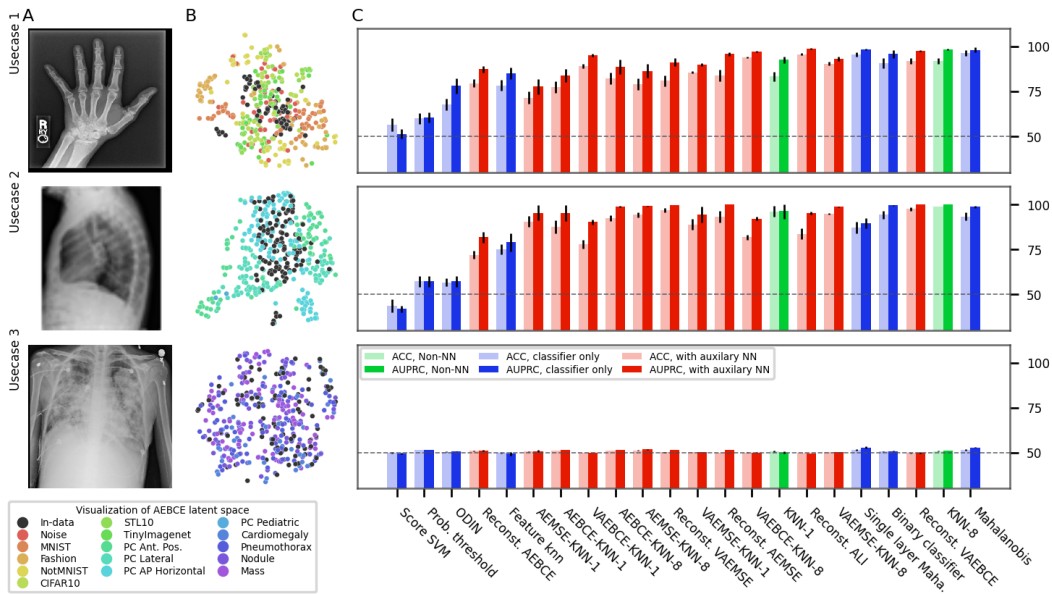

Figure 3: Visualizations and OoDD results on frontal view chest-xray (Evaluation 1). Each row of figures correspond to a use-case. Column A shows examples of *Out* data for each use-case (hand x-ray, lateral view chest X-ray, and xray of pneumothorax from top to bottom). Column B shows UMAP visualizations of AE latent space - colors of points represent their respective datasets. Column C plots the accuracy and AUPRC of OoDD methods in each use-case, averaged across all randomized trials. Bars are sorted by average accuracy across all use-cases, and coloured according to method's grouping: green for baseline image space methods, blue for methods based upon the task specific classifier, and red for methods that use an auxiliary neural network. Error bars represent 95% confidence interval.

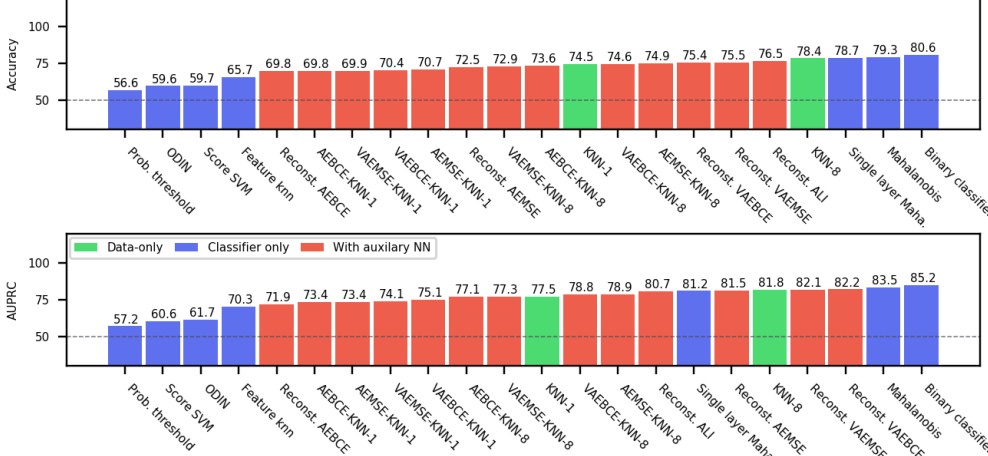

Figure 4: Accuracy and AUPRC of OoDD methods aggregated over all evaluations. Sorted by accuracy from left to right.

across all repeated trials. This illustrates the extreme difficulty of detecting unseen/nouveau diseases, which corroborates the findings of [33].

## 4.1 Overall Performance

Across evaluations, the better performing classifier-only methods are competitive with the methods that use auxiliary models. When performance is aggregated across all evaluations, in Figure 4, the best classifier-only methods (Mahalanobis and binary classifier) outperform auxiliary models in accuracy. The performance of binary classifier is strong despite the method's simplicity. We suspect that this strong performance is due to the fact that we randomly sample 3 *Out* datasets when constructing $D_{val}$ as opposed to selecting a single *Out* dataset. This added variety in $D_{val}$ *Out* data improves generalization by enforcing more stable decision boundaries. We performed additional experiments

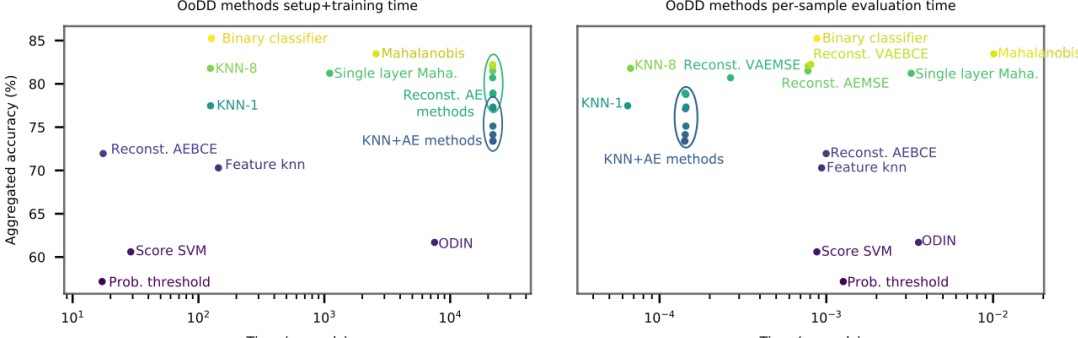

Figure 5: Overall accuracy of methods plotted over total setup time (left) and per-sample run time (right).

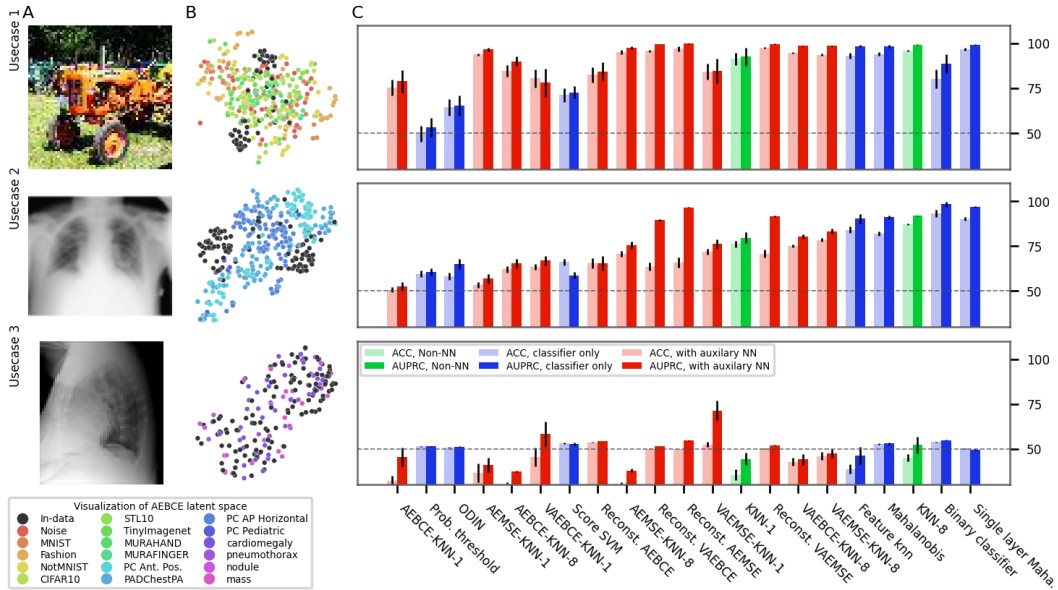

Figure 6: Lateral X-ray imaging (see Figure 3 for description).

with fewer *Out* datasets on a subset of methods and tasks. Results in appendix figure 9 shows that the gap between the top-4 methods quickly closing with more *Out* datasets in $D_{val}$.

## 4.2 Computational Cost

We consider computational cost of each method in terms of setup time and run time in order to add another dimension to compare methods which achieve similar accuracy. The setup time is measured as the wall-clock computation time taken for hyperparameter search and training. For methods with auxiliary models, the training time of auxiliary neural networks are also included in the setup-time. Run time is measured as the per-sample computation time (averaged over fixed batch size) at test time. Figure 5 plots the accuracy of models over their respective setup and run time. All methods can make predictions reasonably fast, allowing for potential online usage. Mahalanobis and its single layer variant take significantly more time to setup and run than other classifier methods. KNN-8 exhibits the best time vs performance trade-off with its low setup time and good performance. However, as it requires the storage of training images for predictions, it may be unsuitable for use on memory constrained platforms (e.g. mobile) or when training data privacy is of concern.

## 5 Discussion

The necessity of OoDD is supported by two considerations. First of which is usability. As we transition ML tools from research labs to the hands of the end user, usability of these tools becomes pivotal to their success. One common characteristic of good usability is to fail gracefully when

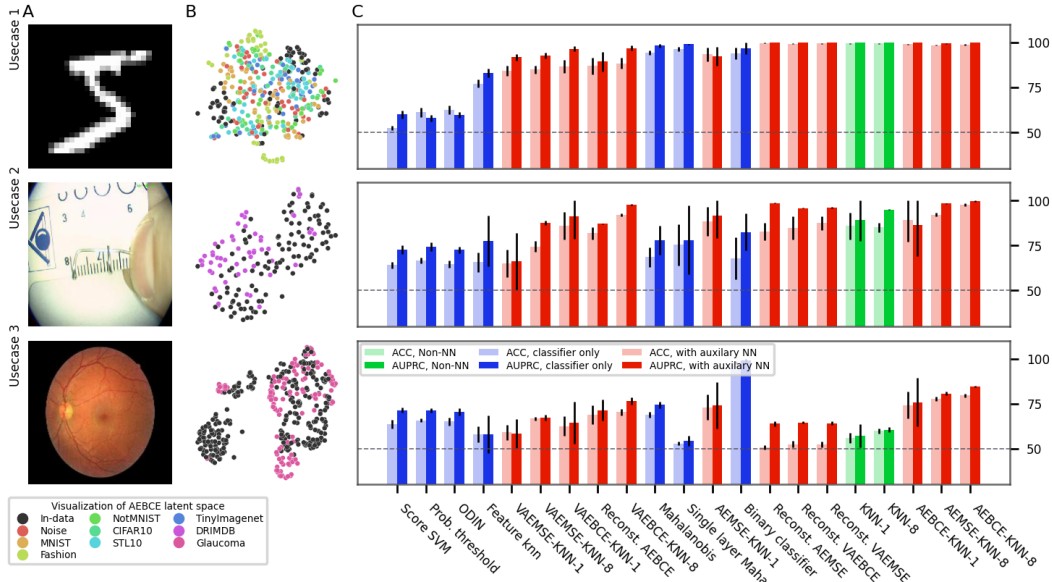

Figure 7: Fundus Imaging (see Figure 3 for description).

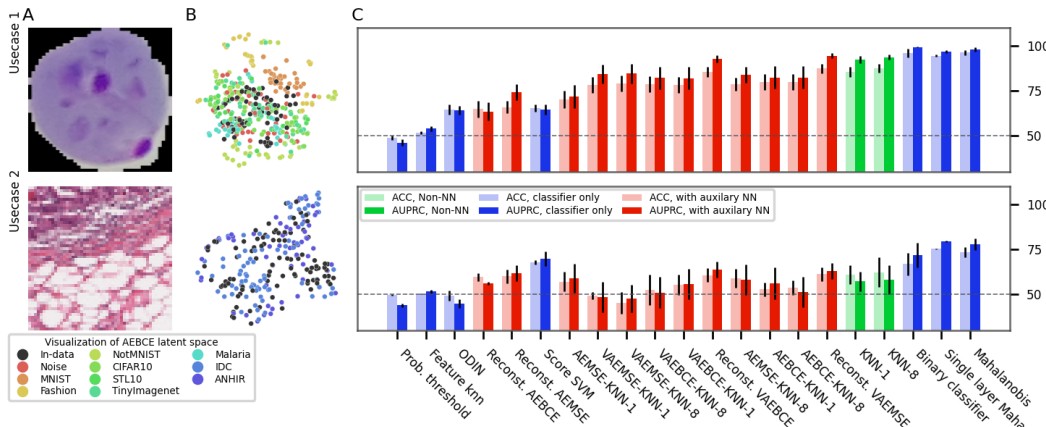

Figure 8: Histology Imaging (see Figure 3 for description).

handling user errors. In ML assisted diagnostic tools, this means equipping the tool with the capacity to reject predictions on erroneous input data, thereby preventing the "garbage-in, garbage-out" scenario. For ML tools facing the general public, this clarity is particularly important. The second reason why OoDD is necessary is the requirement for safety. In applications like ML assisted diagnosis, the performance of the system is directly tied to the safety of the patients. A well documented failure mode for machine learned predictors is when the predictor attempts to extrapolate on inputs outside the distribution of its training data. OoDD provides a safety mechanism that prevents failures of the predictor from harming the user through inaccurate predictions.

## 6 Conclusion

Overall, the top three classifier-only methods obtain better accuracy than all methods with auxiliary models except for fundus imaging. Binary classifier has the best accuracy and AUPRC on average, and is simple to implement. Hence, we recommend binary classifier as the default method for OoDD in the domain of medical images. The methods we find to work best are almost opposite that of [34] despite using the same code for overlapping methods. The main difference between these studies is that they evaluate on natural images instead of medical images. We performed an extensive hyperparameter search on all methods and conclude that this discrepancy is due to the specific data and tasks we have defined. While use-case 1 and 2 are easily solved with non-complicated models, the failure of most models in almost all tasks to significantly solve use-case 3 is consistent with the finding of [35]. Users of diagnostic tools which employ these OoDD methods should still remain

vigilant that images very close to the training distribution yet not in it (and a false negative for use-case 3) could yield unexpected results. In the absence of OoDD methods which have good performance on use-case 3, another approach is to develop methods which will systematically generalize to these examples.

## 7 Limitations

Since we use the downstream task of classifying healthy vs non-healthy for all evaluations, this limits our conclusion to this setting. Other vision tasks such as multiclass classification may provide more useful features and thus see a shift in performance for classifier-based OoDD methods [36]. Furthermore, the *In* and *Out* datasets used span many image domains common to medical imaging, but might not be exactly the challenges faced. While we do not intend our selection of datasets to be exhaustive, we justify the choice of the *Out* data by enumerating different types of mistakes or biases that can occur at different stages of the data acquisition, which we refer to as the *uses-cases*. We kept the same network architecture across experiments; future work may study the effect of the choice of architecture on OoDD performance.

## 8 Related Works

As our focus is on empirically evaluating the performance of OoDD methods in the medical image domain, we refer readers to other review articles [37, 38] for in-depth discussion and meta-analysis of OoDD methods. Our work is also related to other benchmarks on out-of-distribution detection. Domingues et al. [39] surveyed a large number of unsupervised learning algorithms for outlier detection in various data domains. Their formulation of outliers is similar to OoD of use-case 3 in our definition. In contrast to [39], our data is in the image domain, which necessitates our selection of different methods. More recently, Steinbuss et al. [40] proposed to use statistical models to synthetically generate (comparatively low dimensional) outlier data for benchmarking OoDD methods, in order to isolate different types of outliers. Although it would be difficult to scale their method for generating synthetic examples to high resolution images, their proposed framework could provide accurate characterization of OoDD performance in each use-case. Our work is most closely related to [34], which benchmarks a large number of OoDD methods on natural image data. Similar to [34], we also recognize the issue that calibrating and testing OoDD methods on the same *Out* dataset overestimates their performance at generalizing to unknown outliers. Our approach differs in that we use multiple disjoint datasets for calibration and testing where possible to better simulate real world scenarios. Predictive uncertainty modelling is an adjacent task to OoDD that also aims to improve the reliability of ML systems. Ovadia et al. [41] evaluates the predictive uncertainty of deep probabilistic models on OoD samples and finds that the quality of uncertainty modelling degrades with domain shift. This suggests that OoDD methods based solely on predictive uncertainty (e.g. probability threshold) are unlikely to be successful, which is in agreement to our findings. To our best knowledge, we are the first benchmark for OoDD in the medical image domain.

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

# A  Details of Experimental Procedure

The code used for all experiments is provided here: https://github.com/caotians1/OD-test-master

## A.1  Network training

For classifier models, we use a DenseNet-121 architecture [42] with Imagenet pretrained weights. The last layer is re-initialized and the full network is finetuned on $D_{tr}$. As the NIH and PC-Lateral datasets only contain grayscale images, the pretrained weights of features in the first layer are averaged across channels prior to finetuning.

For all of the autoencoders, we use a 12-layer CNN architecture with a bottleneck dimension of 512 for all evaluations. Due to computational constraints, all images are downsampled to $64 \times 64$ when fed to an autoencoder. These AEs are trained from scratch on their respective $D_{tr}$ with MSE loss and BCE loss. We also trained VAEs with the same architectures, except that the bottleneck dimension is doubled to 1024 to allow the code to be split into means and variances.

In addition, we explore the potential benefits of training encoder+decoder using ALI in evaluation 1. We use the same network architecture as proposed in [16], with weights pretrained on Imagenet and finetuned on NIH *In* classes. Due to the added complexity of training GANs and the lack of significant improvements in OoDD performance over regular AEs (see §4), we did not train ALI models for the other three evaluations.

In order to gauge training progress and overfitting, we hold out 5% of $D_{tr}$ as validation set. We select the training checkpoint with the lowest error on $D_{tr}$ for use in OoDD methods.

## A.2  OoDD Method Training

When training the OoDD methods for use-case 1, three *Out* datasets are randomly selected for $D_{val}$ while the rest is used for $D_{test}$. For use-cases 2 and 3, we enumerate over configurations where each *Out* dataset is used as $D_{val}$ with the rest as $D_{test}$. $D_{val}$ and $D_{test}$ are class-balanced by subsampling equal numbers of *In* and *Out* samples. Additionally, some methods (ODIN and Mahalanobis) require additional hyper-parameter selection. Hence, we further subdivide $D_{val}$ in to a 80% 'training' split and a 20% 'validation' split; methods are trained/optimized on the 'training' split with early-stopping/calibration on the 'validation' split. Hyperparameter sweep is carried out where needed. 10 repeated trials, with re-sampled $D_{val}$ and $D_{test}$, are performed for each evaluation.

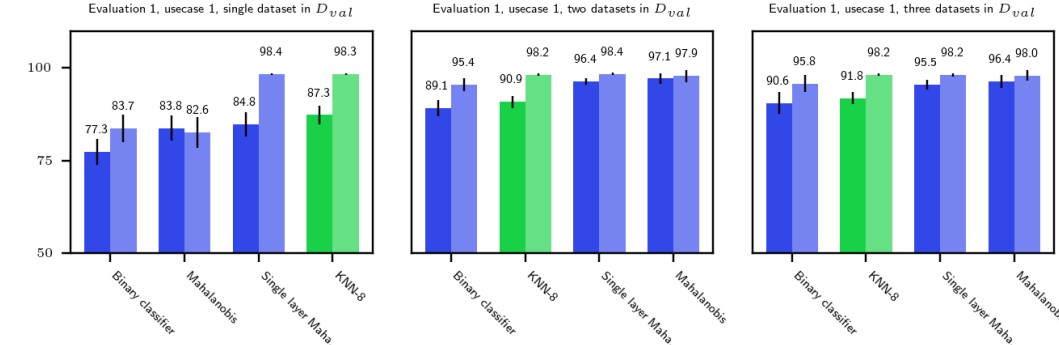

Figure 9: Performance of top-4 methods on frontal X-ray imaging, use-case 1, when trained with fewer datasets in $D_{val}$

| | Usecase 1 | | Usecase 2 | | Usecase 3 | |
|---|---|---|---|---|---|---|
| Method | Acc. (%) | AUPRC (%) | Acc. (%) | AUPRC (%) | Acc. (%) | AUPRC (%) |
| Prob. threshold | $56.4 \pm 3.5$ | $51.2 \pm 2.6$ | $43.6 \pm 3.7$ | $41.9 \pm 1.8$ | $49.8 \pm 0.3$ | $49.5 \pm 0.4$ |
| Score SVM | $59.9 \pm 3.0$ | $60.3 \pm 2.7$ | $57.2 \pm 3.2$ | $57.2 \pm 3.0$ | $51.4 \pm 0.0$ | $51.5 \pm 0.0$ |
| Binary classifier | $67.7 \pm 3.3$ | $78.0 \pm 4.2$ | $56.7 \pm 2.3$ | $57.2 \pm 3.2$ | $50.4 \pm 0.1$ | $50.5 \pm 0.1$ |
| ODIN | $79.5 \pm 2.4$ | $87.2 \pm 1.9$ | $72.0 \pm 2.1$ | $81.9 \pm 3.0$ | $51.0 \pm 0.2$ | $51.0 \pm 0.4$ |
| Mahalanobis | $78.2 \pm 3.1$ | $85.0 \pm 3.2$ | $75.0 \pm 2.8$ | $78.9 \pm 5.1$ | $49.6 \pm 0.2$ | $49.3 \pm 0.9$ |
| Single layer Maha. | $71.4 \pm 3.5$ | $77.7 \pm 4.2$ | $90.6 \pm 3.0$ | $95.4 \pm 4.4$ | $50.5 \pm 0.1$ | $50.7 \pm 0.6$ |
| Feature knn | $77.4 \pm 3.3$ | $83.8 \pm 3.8$ | $87.7 \pm 3.7$ | $95.1 \pm 4.4$ | $51.0 \pm 0.1$ | $51.3 \pm 0.0$ |
| Reconst. AEBCE | $88.9 \pm 1.2$ | $95.1 \pm 0.9$ | $77.9 \pm 2.5$ | $90.2 \pm 1.3$ | $50.0 \pm 0.0$ | $49.9 \pm 0.1$ |
| Reconst. AEMSE | $82.3 \pm 3.2$ | $88.7 \pm 3.9$ | $92.3 \pm 1.6$ | $98.9 \pm 0.3$ | $50.9 \pm 0.1$ | $51.3 \pm 0.0$ |
| Reconst. VAEBCE | $79.0 \pm 3.4$ | $86.3 \pm 3.8$ | $94.2 \pm 1.2$ | $99.2 \pm 0.2$ | $51.2 \pm 0.1$ | $51.7 \pm 0.0$ |
| Reconst. VAEMSE | $80.8 \pm 3.1$ | $91.2 \pm 2.2$ | $96.7 \pm 1.2$ | $99.7 \pm 0.1$ | $50.2 \pm 0.2$ | $51.4 \pm 0.0$ |
| Reconst. ALI | $85.6 \pm 0.7$ | $89.9 \pm 0.9$ | $89.0 \pm 3.0$ | $94.4 \pm 4.3$ | $50.2 \pm 0.0$ | $50.3 \pm 0.0$ |
| KNN-1 | $83.9 \pm 3.2$ | $95.8 \pm 1.1$ | $93.3 \pm 3.4$ | $99.9 \pm 0.0$ | $50.0 \pm 0.0$ | $51.5 \pm 0.0$ |
| KNN-8 | $93.9 \pm 0.5$ | $97.1 \pm 0.5$ | $81.7 \pm 1.4$ | $92.1 \pm 1.1$ | $50.0 \pm 0.0$ | $50.0 \pm 0.1$ |
| VAEMSE-KNN-1 | $83.3 \pm 2.7$ | $92.5 \pm 1.8$ | $96.2 \pm 3.1$ | $96.6 \pm 4.4$ | $50.6 \pm 0.2$ | $50.0 \pm 0.5$ |
| VAEBCE-KNN-1 | $95.7 \pm 0.7$ | $98.7 \pm 0.3$ | $83.8 \pm 3.0$ | $95.2 \pm 0.7$ | $50.0 \pm 0.0$ | $49.5 \pm 0.0$ |
| AEMSE-KNN-1 | $90.4 \pm 1.0$ | $93.2 \pm 1.2$ | $94.7 \pm 0.4$ | $98.8 \pm 0.2$ | $50.3 \pm 0.0$ | $50.3 \pm 0.0$ |
| AEBCE-KNN-1 | $95.5 \pm 1.3$ | $98.2 \pm 0.4$ | $87.2 \pm 3.2$ | $89.6 \pm 2.7$ | $51.6 \pm 0.3$ | $52.8 \pm 0.5$ |
| VAEMSE-KNN-8 | $90.6 \pm 2.9$ | $95.8 \pm 2.3$ | $94.2 \pm 2.3$ | $99.5 \pm 0.3$ | $50.5 \pm 0.1$ | $50.8 \pm 0.1$ |
| VAEBCE-KNN-8 | $91.9 \pm 1.6$ | $97.6 \pm 0.4$ | $97.5 \pm 0.9$ | $99.9 \pm 0.0$ | $49.7 \pm 0.2$ | $50.0 \pm 0.4$ |
| AEMSE-KNN-8 | $91.8 \pm 1.7$ | $98.2 \pm 0.3$ | $98.9 \pm 0.2$ | $99.9 \pm 0.0$ | $50.8 \pm 0.1$ | $51.1 \pm 0.0$ |
| AEBCE-KNN-8 | $96.4 \pm 1.6$ | $98.0 \pm 1.4$ | $93.4 \pm 2.2$ | $98.7 \pm 0.7$ | $51.4 \pm 0.2$ | $52.8 \pm 0.4$ |

Table 2: OoDD performance with NIHCC as *In* data. Error margin reflects standard deviation.

| | Usecase 1 | | Usecase 2 | | Usecase 3 | |
|---|---|---|---|---|---|---|
| Method | Acc. (%) | AUPRC (%) | Acc. (%) | AUPRC (%) | Acc. (%) | AUPRC (%) |
| Prob. threshold | $52.4 \pm 1.2$ | $59.9 \pm 2.2$ | $64.1 \pm 1.9$ | $72.7 \pm 2.3$ | $63.4 \pm 2.5$ | $71.4 \pm 1.5$ |
| Score SVM | $61.1 \pm 2.5$ | $57.9 \pm 1.8$ | $66.6 \pm 1.5$ | $74.3 \pm 2.5$ | $65.7 \pm 0.9$ | $71.2 \pm 1.1$ |
| Binary classifier | $62.5 \pm 2.3$ | $59.6 \pm 1.7$ | $64.5 \pm 1.9$ | $72.5 \pm 1.6$ | $65.0 \pm 2.1$ | $70.3 \pm 1.9$ |
| ODIN | $77.1 \pm 2.2$ | $83.1 \pm 2.3$ | $65.6 \pm 5.3$ | $77.4 \pm 14.1$ | $58.0 \pm 4.5$ | $58.0 \pm 10.4$ |
| Mahalanobis | $84.3 \pm 2.7$ | $91.8 \pm 1.8$ | $65.0 \pm 7.7$ | $66.0 \pm 15.8$ | $59.0 \pm 4.3$ | $58.2 \pm 8.1$ |
| Single layer Maha. | $85.0 \pm 2.2$ | $92.6 \pm 1.7$ | $74.4 \pm 3.0$ | $87.7 \pm 1.1$ | $66.7 \pm 0.9$ | $67.1 \pm 1.6$ |
| Feature knn | $86.6 \pm 3.5$ | $96.5 \pm 1.4$ | $85.9 \pm 7.5$ | $91.1 \pm 12.4$ | $62.5 \pm 5.3$ | $64.3 \pm 11.7$ |
| Reconst. AEBCE | $86.8 \pm 4.5$ | $89.3 \pm 5.4$ | $81.8 \pm 3.3$ | $87.1 \pm 0.2$ | $68.7 \pm 5.4$ | $71.3 \pm 5.8$ |
| Reconst. AEMSE | $88.4 \pm 3.0$ | $96.8 \pm 1.4$ | $91.9 \pm 0.9$ | $97.7 \pm 0.4$ | $70.2 \pm 1.7$ | $76.4 \pm 2.1$ |
| Reconst. VAEBCE | $94.4 \pm 1.2$ | $98.1 \pm 0.9$ | $68.5 \pm 5.4$ | $77.9 \pm 8.2$ | $68.8 \pm 1.6$ | $74.2 \pm 1.9$ |
| Reconst. VAEMSE | $96.3 \pm 1.3$ | $99.0 \pm 0.2$ | $75.4 \pm 11.4$ | $78.1 \pm 19.1$ | $52.8 \pm 1.0$ | $54.1 \pm 2.8$ |
| KNN-1 | $93.3 \pm 3.7$ | $92.4 \pm 5.2$ | $88.5 \pm 8.0$ | $91.7 \pm 12.5$ | $72.9 \pm 7.2$ | $73.9 \pm 12.9$ |
| KNN-8 | $94.0 \pm 3.3$ | $96.8 \pm 3.3$ | $67.8 \pm 11.5$ | $82.5 \pm 10.4$ | $97.7 \pm 0.7$ | $99.0 \pm 0.7$ |
| VAEMSE-KNN-1 | $99.6 \pm 0.2$ | $100.0 \pm 0.0$ | $82.8 \pm 4.7$ | $98.6 \pm 0.2$ | $50.7 \pm 1.2$ | $63.9 \pm 1.3$ |
| VAEBCE-KNN-1 | $99.3 \pm 0.3$ | $100.0 \pm 0.0$ | $84.6 \pm 6.4$ | $95.6 \pm 0.2$ | $52.4 \pm 1.9$ | $64.5 \pm 0.7$ |
| AEMSE-KNN-1 | $99.3 \pm 0.2$ | $100.0 \pm 0.0$ | $87.5 \pm 3.8$ | $96.0 \pm 0.3$ | $52.2 \pm 1.5$ | $64.1 \pm 1.0$ |
| AEBCE-KNN-1 | $99.3 \pm 0.2$ | $100.0 \pm 0.0$ | $85.8 \pm 7.5$ | $89.3 \pm 11.9$ | $55.8 \pm 2.9$ | $57.0 \pm 6.4$ |
| VAEMSE-KNN-8 | $99.4 \pm 0.1$ | $100.0 \pm 0.0$ | $85.0 \pm 2.7$ | $95.0 \pm 0.1$ | $59.7 \pm 1.4$ | $60.5 \pm 1.4$ |
| VAEBCE-KNN-8 | $99.0 \pm 0.3$ | $99.9 \pm 0.0$ | $89.2 \pm 12.2$ | $86.2 \pm 17.1$ | $74.2 \pm 7.5$ | $75.8 \pm 13.4$ |
| AEMSE-KNN-8 | $98.5 \pm 0.3$ | $99.4 \pm 0.2$ | $92.2 \pm 1.1$ | $98.3 \pm 0.1$ | $77.6 \pm 1.1$ | $80.6 \pm 1.0$ |
| AEBCE-KNN-8 | $98.9 \pm 0.3$ | $99.9 \pm 0.0$ | $97.6 \pm 0.9$ | $99.6 \pm 0.5$ | $79.5 \pm 1.0$ | $84.6 \pm 0.5$ |

Table 3: OoDD performance with DRD as *In* data. Error margin reflects standard deviation.

| Method | Usecase 1 Acc. (%) | Usecase 1 AUPRC (%) | Usecase 2 Acc. (%) | Usecase 2 AUPRC (%) | Usecase 3 Acc. (%) | Usecase 3 AUPRC (%) |
|---|---|---|---|---|---|---|
| Prob. threshold | $75.3 \pm 4.5$ | $78.8 \pm 6.4$ | $50.6 \pm 1.5$ | $52.6 \pm 2.3$ | $32.2 \pm 2.8$ | $45.4 \pm 5.3$ |
| Score SVM | $49.7 \pm 4.4$ | $53.2 \pm 5.1$ | $59.5 \pm 1.9$ | $60.6 \pm 2.1$ | $51.3 \pm 0.2$ | $51.3 \pm 0.3$ |
| Binary classifier | $64.3 \pm 4.8$ | $65.1 \pm 5.7$ | $58.2 \pm 2.1$ | $65.0 \pm 2.9$ | $50.8 \pm 0.2$ | $51.1 \pm 0.2$ |
| ODIN | $93.7 \pm 0.6$ | $96.6 \pm 0.9$ | $53.2 \pm 1.5$ | $56.9 \pm 2.6$ | $36.7 \pm 5.3$ | $41.0 \pm 4.0$ |
| Mahalanobis | $84.7 \pm 3.3$ | $90.0 \pm 2.7$ | $62.0 \pm 1.9$ | $65.2 \pm 2.5$ | $29.7 \pm 1.5$ | $37.3 \pm 0.4$ |
| Single layer Maha. | $80.5 \pm 5.1$ | $78.1 \pm 7.8$ | $63.5 \pm 1.6$ | $67.0 \pm 2.6$ | $45.6 \pm 5.3$ | $58.3 \pm 6.9$ |
| Feature knn | $71.1 \pm 3.7$ | $72.7 \pm 3.6$ | $66.2 \pm 1.8$ | $58.7 \pm 1.8$ | $53.1 \pm 0.4$ | $52.7 \pm 0.7$ |
| Reconst. AEBCE | $82.6 \pm 4.2$ | $84.4 \pm 5.0$ | $65.3 \pm 2.8$ | $65.3 \pm 4.0$ | $53.6 \pm 0.2$ | $54.3 \pm 0.2$ |
| Reconst. AEMSE | $95.0 \pm 1.3$ | $97.4 \pm 0.7$ | $70.5 \pm 1.6$ | $75.5 \pm 1.8$ | $28.7 \pm 2.6$ | $37.8 \pm 1.0$ |
| Reconst. VAEBCE | $95.7 \pm 0.6$ | $99.6 \pm 0.1$ | $63.6 \pm 2.1$ | $89.4 \pm 0.5$ | $50.0 \pm 0.0$ | $51.4 \pm 0.2$ |
| Reconst. VAEMSE | $97.0 \pm 1.3$ | $99.8 \pm 0.1$ | $65.8 \pm 3.0$ | $96.5 \pm 0.2$ | $50.0 \pm 0.0$ | $54.6 \pm 0.2$ |
| KNN-1 | $84.2 \pm 4.4$ | $84.7 \pm 6.8$ | $72.0 \pm 1.7$ | $76.1 \pm 2.5$ | $52.4 \pm 1.5$ | $71.2 \pm 5.5$ |
| KNN-8 | $91.3 \pm 3.5$ | $92.5 \pm 5.2$ | $76.1 \pm 1.8$ | $79.7 \pm 3.2$ | $35.4 \pm 3.0$ | $44.4 \pm 3.6$ |
| VAEMSE-KNN-1 | $97.5 \pm 0.5$ | $99.7 \pm 0.1$ | $70.8 \pm 2.1$ | $91.6 \pm 0.3$ | $50.1 \pm 0.1$ | $51.9 \pm 0.2$ |
| VAEBCE-KNN-1 | $94.8 \pm 0.5$ | $98.7 \pm 0.2$ | $75.1 \pm 0.9$ | $80.3 \pm 1.3$ | $42.8 \pm 2.4$ | $44.1 \pm 3.1$ |
| AEMSE-KNN-1 | $93.7 \pm 0.6$ | $98.5 \pm 0.3$ | $78.4 \pm 1.1$ | $83.3 \pm 1.4$ | $45.9 \pm 2.2$ | $47.4 \pm 2.5$ |
| AEBCE-KNN-1 | $93.1 \pm 1.6$ | $98.5 \pm 0.5$ | $84.0 \pm 1.7$ | $90.2 \pm 2.5$ | $38.7 \pm 2.5$ | $46.2 \pm 5.0$ |
| VAEMSE-KNN-8 | $94.1 \pm 1.0$ | $98.3 \pm 0.6$ | $82.0 \pm 1.3$ | $91.0 \pm 0.9$ | $52.6 \pm 0.3$ | $52.9 \pm 0.5$ |
| VAEBCE-KNN-8 | $95.8 \pm 0.4$ | $99.3 \pm 0.2$ | $87.2 \pm 0.5$ | $91.9 \pm 0.6$ | $45.0 \pm 2.0$ | $52.2 \pm 4.7$ |
| AEMSE-KNN-8 | $80.2 \pm 5.3$ | $88.7 \pm 5.3$ | $93.1 \pm 2.1$ | $98.3 \pm 1.5$ | $53.8 \pm 0.2$ | $54.9 \pm 0.2$ |
| AEBCE-KNN-8 | $96.7 \pm 0.7$ | $99.3 \pm 0.2$ | $90.0 \pm 1.0$ | $96.8 \pm 0.3$ | $50.0 \pm 0.2$ | $49.2 \pm 0.4$ |

Table 4: OoDD performance with PadChest as *In* data. Error margin reflects standard deviation.

| Method | Usecase 1 Acc. (%) | Usecase 1 AUPRC (%) | Usecase 2 Acc. (%) | Usecase 2 AUPRC (%) |
|---|---|---|---|---|
| Prob. threshold | $49.0 \pm 1.1$ | $46.3 \pm 1.9$ | $49.7 \pm 0.4$ | $43.9 \pm 0.9$ |
| Score SVM | $51.8 \pm 0.8$ | $54.1 \pm 1.4$ | $50.1 \pm 0.1$ | $51.5 \pm 1.0$ |
| Binary classifier | $64.7 \pm 2.8$ | $64.1 \pm 2.4$ | $49.0 \pm 2.8$ | $44.7 \pm 2.4$ |
| ODIN | $64.9 \pm 4.5$ | $63.6 \pm 5.0$ | $59.5 \pm 2.1$ | $56.0 \pm 0.9$ |
| Mahalanobis | $66.0 \pm 3.4$ | $74.4 \pm 4.4$ | $59.9 \pm 4.0$ | $61.6 \pm 4.4$ |
| Single layer Maha. | $65.5 \pm 1.9$ | $64.8 \pm 2.8$ | $67.8 \pm 1.1$ | $69.6 \pm 4.1$ |
| Feature knn | $70.3 \pm 4.7$ | $71.9 \pm 6.4$ | $57.0 \pm 5.4$ | $58.8 \pm 8.2$ |
| Reconst. AEBCE | $78.4 \pm 4.3$ | $84.5 \pm 5.2$ | $49.2 \pm 2.0$ | $48.2 \pm 8.4$ |
| Reconst. AEMSE | $79.0 \pm 4.4$ | $84.6 \pm 5.4$ | $45.0 \pm 6.1$ | $47.6 \pm 7.8$ |
| Reconst. VAEBCE | $78.8 \pm 4.4$ | $82.4 \pm 6.3$ | $52.4 \pm 8.4$ | $50.9 \pm 8.7$ |
| Reconst. VAEMSE | $78.5 \pm 4.4$ | $82.2 \pm 6.2$ | $55.1 \pm 5.7$ | $55.6 \pm 8.4$ |
| KNN-1 | $85.6 \pm 2.9$ | $92.9 \pm 1.9$ | $60.7 \pm 3.7$ | $63.7 \pm 4.5$ |
| KNN-8 | $78.8 \pm 3.5$ | $84.1 \pm 4.5$ | $59.0 \pm 5.3$ | $58.2 \pm 8.4$ |
| VAEMSE-KNN-1 | $80.1 \pm 4.4$ | $82.5 \pm 6.3$ | $52.6 \pm 3.8$ | $56.0 \pm 8.9$ |
| VAEBCE-KNN-1 | $80.1 \pm 4.4$ | $82.6 \pm 6.3$ | $53.5 \pm 4.0$ | $51.0 \pm 8.4$ |
| AEMSE-KNN-1 | $87.5 \pm 2.5$ | $94.6 \pm 1.5$ | $61.1 \pm 3.8$ | $62.9 \pm 4.3$ |
| AEBCE-KNN-1 | $85.6 \pm 2.9$ | $92.4 \pm 2.1$ | $60.9 \pm 5.1$ | $57.1 \pm 5.5$ |
| VAEMSE-KNN-8 | $87.6 \pm 2.4$ | $93.9 \pm 1.5$ | $62.3 \pm 8.1$ | $57.9 \pm 8.1$ |
| VAEBCE-KNN-8 | $96.1 \pm 2.5$ | $99.5 \pm 0.3$ | $66.7 \pm 6.3$ | $71.8 \pm 6.9$ |
| AEMSE-KNN-8 | $94.6 \pm 0.5$ | $96.8 \pm 0.8$ | $75.2 \pm 0.3$ | $79.4 \pm 0.5$ |
| AEBCE-KNN-8 | $96.4 \pm 1.3$ | $98.2 \pm 1.3$ | $73.5 \pm 2.8$ | $77.7 \pm 3.2$ |

Table 5: OoDD performance with PCAM as *In* data. Error margin reflects standard deviation.

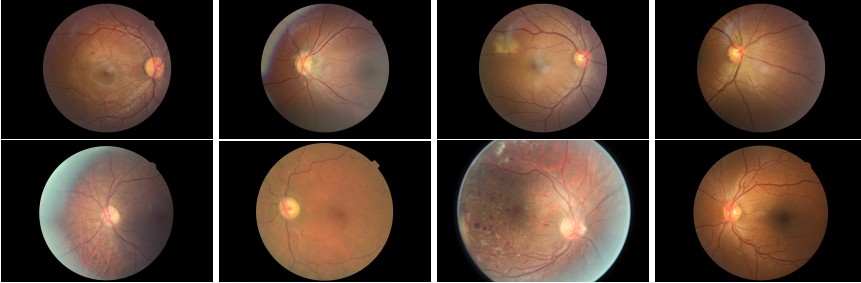

Figure 10: Comparison of RIGA and DRD images: Top row are images sampled from RIGA, while bottom row are images sampled from DRD. There are notable visual differences between glaucoma and diabetic retinopathy.

