# OpenReview forum: "A Benchmark of Medical Out of Distribution Detection"
_NeurIPS.cc/2021/Track/Datasets_and_Benchmarks/Round1 — Submitted to NeurIPS 2021 Datasets and Benchmarks Track (Round 1)_

### Official Review · Reviewer_BXxy · 2021-06-30
**OoDD Medical**

**Rating:** 6
**Confidence:** 3
**Correctness:** 1. Please explain why AUPRC is a good…
**Clarity:** 1. The large number of datasets and m…

**Strengths:**

1. Wide variety and a large number of evaluated datasets.
2. Wide variety in the techniques evaluated.
3. This work attempts to baseline a real problem concerning ML in medical imaging.
4. The use of different cases helps qualify the problem of out of distribution detection in a meaningful way.

**Weaknesses:**

1. I'd prefer more details to better understand the statement on line 68. Why? Is this captured in the standard error from line 76?
2. Many of the bar diagrams are difficult to compare? How are they ordered on the x-axis?
3. The methods section (3.1) does very little to explain the algorithms used for evaluation. Please expand on the information of at least the top-performing methods in each group.

**Additional Feedback:**

Use case #3 is closer to a typical detection problem than OoDD from my point of view. For this reason, it makes sense that performance was very low. What could be done to improve results in this area?

I believe all the technical work is done to make a stronger paper. I'd recommend focusing on clarity in the diagrams. I like the use case grouping as it allows the reader to clearly see differences. Can a cleaner grouping be applied to the methods? What conclusions can be drawn from the large number of tests conducted? What was surprising and how can that be explained.

**Documentation:**

Datasets are referenced and linked via a git site

**Ethics:**

I did not identify any concerns

**Relation To Prior Work:**

Yes, technically similar to previous methods for OoDD but this work specifically test medical datasets

**Summary And Contributions:**

This paper attempts to benchmark multiple methods for identifying test images outside the distribution of the training data in the field of medical imaging. The authors compile 17 different datasets in order to draw conclusions for a wide range of applications. This paper compiles metrics for accuracy as well as computational cost. The overall results show that classification for use cases #1 and #2 are possible, but use case #3 had a poor performance for all tested methods.

---

### Official Review · Reviewer_NH16 · 2021-07-04
**Review comments (updated 19 Jul.)**

**Rating:** 6
**Confidence:** 3
**Clarity:** Yes, the paper is generally well writ…

**Strengths:**

+ The analysis of out of distribution is important for medical imaging, and the provided benchmark would benefit some research studies as well as clinical applications.

+ The running example is helpful for understanding the proposed scheme.

+ The finding from the experimental analysis is interesting and could be used to help some clinical applications. The suggestion of using a simple binary classifier is also helpful for the guidance of applying machine learning models to clinical applications.

+ The paper is generally well-written.

**Weaknesses:**

- The authors made clear definitions of three Out of Distribution (OoD) data (Sec. 2). But it lacks evidence to support the definition. Why it has to be defined in this way. Please provide either reference, clinical findings or other justifications to support the definition.

- The task formulation (Sec. 3) is a bit unclear. The authors started by talking about "downstream task". Then what is the upstream/pre-training task?
Taking the classifier-based methods as an example, if the way they are used for detecting OoD data is to directly use the threshold of prediction or logits, the training of the binary classifier sounds more like an upstream task.
It is also a bit confusing why the "methods with auxiliary models" requires pre-training of a network like image reconstruction. The auxiliary task is already the prediction of In/Out of distribution, then why there has to be a pre-training step?

- When preparing the In data, the authors mentioned that they "randomly split the dataset" into train/val/test (for some of the datasets used). But it is unclear how this random process was performed. Are the identity of different patients taken into account, or just randomly split over the whole dataset? If it is the latter case, there might be the risk of data leakage.

- It is unclear if the same network architecture/backbone is used across all the experimental evaluations. If so, it would be interesting to see how different networks perform, especially for the use-case 3 where the authors owe the poor performance to the difficulty of detecting the OoD data in this case. If not, then the evaluation is not fair and the conclusion could be misleading then.

- It is a bit unclear why to present the computational cost (in time) for the benchmark (Sec. 4.2). I know it is always good to have more metrics when benchmarking methods, but the setup/run time here is not well motivated. If the application to mobile devices, the size of the model (e.g. the number of parameters) and the memory consumption could also be included for comparison but were not.  Please clarify the reason and at least provide a convincing motivation at the beginning of this subsection.

- The authors pointed one limitation of this work at the end of the paper, i.e. using "classifying healthy vs non-healthy" for evaluation. Then why not using the multiclass classification task? Usually the datasets used have corresponding labels to support such a task and it is not difficult to extend the binary classification framework to a multi-class one. It would make the paper and the conclusion more complete and convincing by including this task. Please provide a convincing reason for not doing this experiment.

- There lacks a section of discussion to related work and the difference between them.

- Missing definition for the term "ODIN" (L100).

**Additional Feedback:**

* The performance of different methods as shown in the bar charts in Fig. 3, 6, 7, 8 are very close. It would be better to visualize the performance in another way or just use a table to present. If the authors intend to use the bar chart, suggest adding the numerical results to the bars for a more clear comparison.

* For the use-case 1 datasets, the authors choose some datasets consisting of natural images. It would be interesting to see how the benchmark looks like if this use-case includes some other images like satellite images, sketches, depth images, etc. I believe this would make the conclusion stronger.

* I would suggest revising Fig. 2 to make it more clear. I assume the figure intends to illustrate the two main "downstream tasks", but the figure in its current form is a bit confusing. It would be better to re-arrange it into two pipelines of the corresponding tasks.

**Correctness:**

The claims made are mostly correct. Some of the evaluation methods and experimental design are unclear, making it not easy to assess their correctness, e.g. the definition of the task formulation, the data split and the network architecture used.

**Documentation:**

There lack sufficient details (as mentioned in the above sections) to reproduce exactly the same results purely based on the descriptions provided in the paper. But since the code (link) is provided, I believe it would support reproducibility.

**Ethics:**

The datasets used in this paper are all from existing publicly available datasets, there should not be ethical concerns.

**Relation To Prior Work:**

There is not a part/section discussing how this work differs from previous contributions and related works. Only some sentences mentioned a bit, but lacks a thorough overview and discussion.

**Summary And Contributions:**

In this paper, the authors present a benchmark of out of distribution detection (OoDD) for medical data. Specifically, they define three categories of OoD examples and benchmark several OoDD methods across different medical imaging data (including chest X-ray, fundus imaging and histology slides). Experiments show that some well-performed methods (on some categories) fail to recognize images close to the training distribution, and a simple binary classifier performs the best among the evaluated methods.

The main contribution of this paper is its extensive experimental evaluations of different methods on several medical datasets with different settings. The conclusion drawn from the experiments is also interesting and could possibly guide some clinical applications.

I have read the rebuttal and more details please refer to my response below.

---

### Official Review · Reviewer_NTH1 · 2021-07-07
**A Benchmark of Medical Out of Distribution Detection (updated 20th Jul.)**

**Rating:** 6
**Confidence:** 4
**Clarity:** The paper requires a major revision o…

**Strengths:**

Authors tackle an important problem in medical imaging: developing confident solutions to new cases in health applications. Therefore understanding the behavior of available techniques is important in this domain. Authors applied a comprehensive study in this regard.

**Weaknesses:**

Although the study is comprehensive, still lacks some important techniques, for instance metric learning. The UMAP results show there is no clear boundary among seen and unseen samples even in the easiest use-case 1.

Also, the organization of the material requires a major revision. Besides I miss a comprehensive discussion on the graphics.

**Additional Feedback:**

The abstract style of writing does not look standard to me.

**Correctness:**

The experiments are correct. However, they need a better organization in the text.

**Documentation:**

There are many methods in the comparison and hyperparameters and training details are not sufficiently explained for reproducibility.

**Ethics:**

I guess there is no ethical concerns in this work.

**Relation To Prior Work:**

I miss some history on the related works at the beginning of the paper.

**Summary And Contributions:**

Authors perform a study on several available techniques to detect out-of-distribution samples in different setups on medical images. As a result, a binary classifier is the winner in a compromise between accuracy and time.

I've read authors' response. Although I'm not convinced with the answers, I keep my current rating.

---

### Decision · Program_Chairs · 2021-07-26

**Decision:**

Reject

**Comment:**

The problem presented in the paper and its evaluation is very relevant to the community. However, the major concern is the limited contribution in terms of new fresh data. The paper defines an evaluation based on a large set of existing datasets, which is very valuable and comprehensive, but limits the novelty/added value regarding the data contribution point of view.